# Does controlling shareholders' share pledge exacerbate excessive financialization of enterprises?—Evidence from performance pressure perspective

**Yue Xie, Tianhui Wang*****, Jinhua Zhang, Na Wang**

School of Economics, Zhejiang University of Technology, Hangzhou, China

* wang_1999_1@163.com

**Data Availability Statement:** All relevant data are within the paper and its Supporting information files.

## Abstract

Based on the perspective of performance pressure, we explore the influence of controlling shareholders' share pledge on excessive financialization behavior of enterprises and its internal mechanism. The results show that the share pledge of controlling shareholders is positively correlated with the excessive financialization behavior of enterprises. After the controlling shareholder's share pledge, the actual performance of the enterprise is lower than expected, causing the short-sighted behavior of the management, which makes the management willing to conspire with the controlling shareholder to cause the excessive financialization of the enterprise. The results are especially evident among the uncertainty of economic policy is low, the industry competition is not fierce and the executives have overseas experience.

## Introduction

In recent years, with the downward pressure of the domestic and foreign economies, China's economy has entered a new normal, with the problems of overcapacity and the decline of main business profits in the real enterprises. The economy has shown a significant trend of "turning from real to virtual", and the financialization of non-financial enterprises is its microcosmic manifestation. The behavior of non-financial enterprises' financialization should be divided into two parts. On the one hand, the moderate financialization behavior motivated by capital management can have a positive impact on enterprises. Entity enterprises use the idle funds of enterprises to allocate financial assets to obtain high returns and increase the liquidity of enterprise assets. When the development of enterprises needs funds in the future, the high liquidity of financial assets can meet the capital needs of enterprises in time, as to promote the development of the main business of entity enterprises [1]. On the other hand, excessive financialization motivated by profit-seeking will make enterprises lack sufficient funds for equipment upgrading and product research and development, "crowding out" the industrial investment of enterprises [2], and ultimately inhibit the development of the main business of enterprises [3]. The real economy is the foundation of a country's development. When a large of real enterprises reduce their real investment and use resources

**Funding:** This research was funded by the National Natural Science Foundation of China (grant number: 72002201), URL of funder website: https://www.nsfc.gov.cn/; And the National Social Science Fund of China (grant number: 21BJY256), URL of funder website: http://fz.people.com.cn/skygb/sk/index.php/index/seach/. The funders had no role in study design, data collection and analysis, decision to publish, or preparation of the manuscript.

**Competing interests:** The authors have declared that no competing interests exist.

excessively for financial asset investment, it will cause the formation of a foam in the price of the capital market, which is not conducive to the long-term development of the economy. Therefore, it is of great practical significance to study the influencing factors of excessive financialization and put forward some targeted suggestions to guide real enterprises to "get out of virtual and into real".

Equity pledge is a commitment established by shareholders of listed companies in exchange for funds from banks or other financial institutions. Compared with the traditional mortgage method, the significant advantages of equity pledge are given as follows. On the one hand, shareholders still have the original control right while getting the funds; On the other hand, equity pledge does not require the consent of all shareholders and the approval of the regulatory authority, which is more convenient and easier to operate in terms of guarantee setting. Therefore, equity pledge has gradually become a common means of shareholder financing. After the equity pledge of the controlling shareholder, if the share price falls to the closing line and the shareholder fails to increase the margin in time, the shareholder will face the risk of control transfer. In order to avoid the risk of control transfer, the controlling shareholders will intervene through negative measures such as earnings management and information management to prevent the stock price from falling beyond expectations [4]. At the same time, controlling shareholders can urge the company to increase the allocation of high-yield financial assets by influencing the management's business decisions, in order to improve the company's performance and maintain the stability of the stock price. Unlike the entity investment with a long return cycle and uncertain return, the return of financial asset investment is higher. Increasing financial asset investment can improve the company's profit level in the short term. However, excessive financialization will inhibit the development of the entity enterprise's main business and overdraw the long-term development of the enterprise [3]. Therefore, it is of great theoretical and practical significance to study whether equity pledge of controlling shareholders will cause the excessive investment in financial assets of enterprises. We perform the empirical investigation based on firms listed in the Chinese non-financial, and we first regresses and fits the normal level of corporate financialization, then obtains the degree of corporate excessive financialization, and uses the two-way fixed effect model to empirically test the impact of equity pledge of controlling shareholders on corporate excessive financialization.

We use Chinese data for empirical analysis for two reasons. Firstly, equity pledge transactions are common and relatively large in the Chinese environment compared with other markets. According to the statistics of Wind database, as of January 1, 2023, the pledged market value of China's A-share market was 3.2 trillion yuan, and the number of pledged shares reached 400,836 billion shares, accounting for 5.22% of the total A-share capital. In addition, China's better disclosure system for equity pledge information requires companies to disclose more details to the market in a timely manner, which allows us to accurately identify the start and end dates of each pledge transaction. Secondly, in recent years, China's economy has shown a remarkable trend of "turning from real to virtual". From the perspective of typical cases of financialization, Youngor's total profit in the 22 years from its listing in 1998 to 2020 is about 58 billion yuan, and investment income such as stock trading has contributed about 40 billion yuan. The proportion of profit obtained by manufacturing industry and investment profit such as stock trading is roughly 3:7. Nearly half of Yunnan Baiyao's net profit of 5.5 billion yuan in 2020 came from equity investments. In conclusion, China provides a suitable environment to test the hypothesis that equity pledge of controlling shareholders influences excessive financialization of enterprises.

Compared with the existing research results, the main contributions of this paper are given as follows. (1) Although there have been studies on the relationship between the equity pledge

of controlling shareholders and enterprise financialization, they all focus on the financialization of enterprises, and did not separate moderate financialization from excessive financialization. For example, Du Yong et al. (2021) found an inverted U-shaped relationship between the shareholding pledge ratio of controlling shareholders and the financialization of entity enterprises [5]. Xiong Lihui et al. (2021) believe that equity pledge of controlling shareholders will intensify the degree of financialization of enterprises by intensifying the financing constraints of enterprises [6]. Based on the perspective of catering theory, Wang Haifang et al. (2021) found that the financialization trend of entity enterprises will increase significantly after controlling shareholders' equity pledge, and the catering effect generated by investor sentiment is an important way for controlling shareholders' equity pledge to affect the financialization of entity enterprises [7]. Compared with moderate financialization, excessive financialization of enterprises may be the real incentive for the real economy to "move from real to virtual". Different from the above study, this paper distinguishes the difference between moderate and over-financialized enterprises, and finds that the adverse impact of equity pledge of controlling shareholders on the future performance of enterprises' main businesses only exists in the over-financialized enterprise sample. Therefore, this paper will focus on the excessive financialization of enterprises, as to better understand the impact of equity pledge of controlling shareholders on enterprise financialization. (2) This paper enriched the mechanism and heterogeneity analysis of controlling shareholders' equity pledge on excessive financialization. We introduced performance pressure as a mechanism in the study of the influence of equity pledge of controlling shareholders on excessive financialization of enterprises, explored a new research perspective on the correlation between equity pledge and excessive financialization of enterprises, and explored the heterogeneity from three aspects: economic policy uncertainty, industry competition intensity and overseas experience of senior executives. It has enriched the relevant literature on the influence path of equity pledge and excessive financialization behavior of entity enterprises. (3) By linking the excessive financialization behavior of entity enterprises under the background of controlling shareholders' equity pledge with the future main business performance of enterprises, it provides a useful supplement to the research on the consequences of corporate decisions under the background of controlling shareholders' equity pledge.

## Literature review

A large of documents have studied the economic consequences of equity pledge. After the pledge of equity, the behaviors of controlling shareholders such as occupying funds will damage the company's performance [8] and aggravate the company's financing constraints [9], which will have a negative impact on the innovation investment and innovation efficiency of listed companies [10]. After the pledge of shares, the decline of share price is an important reason for the loss of control of the controlling shareholders. In order to avoid the decline of share price and the transfer of control, the controlling shareholders tend to share buybacks, earnings management, and reduce R&D expenditure to mitigate the risk of control transfer [4, 8, 11], and are more likely to timely disclose good news for the company and hide bad news [12]. However, the above negative measures cannot solve the problem from the root cause. After the pledge is released, the risk of stock price collapse will increase [13]. From the above documents, we can find that in order to stabilize the stock price, the controlling shareholders during the period of equity pledge have the motivation to intervene in the company's operation, as to release a positive signal to the market. Since financial assets have high returns and strong liquidity, and are a relatively hidden means of market value management, after equity pledge, whether the controlling shareholders will join the management to increase the share

price by increasing a large amount of financial assets investment, leading to the risk of excessive financialization of enterprises, which needs further research.

In recent years, corporate financialization has become the focus of the government and academia. Studying corporate financialization is of great significance for guiding finance to better serve the real economy. At present, many studies have explored the influencing factors of enterprise financialization. At the level of micro corporate governance, short-sighted managers, more institutional investors' shareholding, and enterprise business diversification will promote the financial investment of enterprises [14–16], while a better level of corporate governance can inhibit the financial investment of enterprises [17]. At the macro policy environment level, the uncertainty of economic policy will affect the financial investment behavior of enterprises [18, 19], but the impact direction is still controversial. The uncertainty of Climate policy, the introduction of local industrial policies and the improvement of fiscal transparency can inhibit the enterprise financialization [20–22], while the margin trading system will intensify the enterprise financialization [23]. By combing relevant literature, it can be found that most of the existing literature directly defines financialization as excessive, without considering the heterogeneity between moderate and excessive financialization. Based on the above analysis, this paper obtains the excessive financialization level of sample enterprises according to the normal financialization level of each enterprise fitted by OLS regression. On this basis, it explores the impact of equity pledge of controlling shareholders on the excessive financialization level of entity enterprises and its internal mechanism.

## Theoretical analysis and research hypothesis

### Equity Pledge and excessive financialization

Although equity pledge is the personal financial behavior of shareholders, due to the special status of controlling shareholders in the enterprise, when they pledge equity, it may have a certain impact on the performance and risk of listed companies, and then change the company's asset allocation. Firstly, controlling shareholders' share pledge will aggravate the agency problem between controlling shareholders and minority shareholders. After the equity pledge, the original control right of the controlling shareholder has not changed, but the cash flow right during the pledge period belongs to the pledgee. Therefore, the equity pledge leads to the separation of control and cash flow rights, reducing the cost of the controlling shareholder's expropriate wealth from minority shareholders [24]. The exacerbation of the agency problem drives the controlling shareholders to prefer projects that can bring private benefits to them [10], and high-yield financial assets may become their means. The reasons include the following two aspects. On the one hand, the controlling shareholders who still have control rights can obtain high returns through financial investment in a short time, and use related party transactions to transfer profits and hollow out the company. On the other hand, the high return of financial assets is accompanied by high risks. Overinvestment in financial assets is not conducive to the long-term development of the enterprise. The minority shareholders cannot enjoy the dividends brought by the long-term value appreciation of the enterprise, while the controlling shareholders loses part of the cash flow right due to the pledge of equity and is less affected. Therefore, after the equity pledge, the controlling shareholders may reach the maximum profit by allocating more financial assets, thus exacerbating the degree of excessive financialization of enterprises.

Secondly, the equity pledge of controlling shareholders will cause the risk of transfer of corporate control. The equity pledge of the controlling shareholder aggravates the information asymmetry [13]. Investors can't timely understand the intention of the pledge, and may

regard this behavior as negative information that shareholders are facing financial constraints, which often causes irrational panic among investors. If the share price of a listed company falls to the warning line during the period of equity pledge, which causes market panic, it will increase the risk of stock price collapse and bring about the risk of control transfer. Therefore, controlling shareholders have strong incentives to intervene in the management decisions of listed companies to support the price of their pledged shares for their own purpose of maintaining control rights [25]. The relatively centralized ownership structure of Chinese listed companies means that the controlling shareholders have ability to intervene in the company's asset allocation [26], which provides conditions for the controlling shareholders to use financial assets to maintain the stock price. On the one hand, the controlling shareholders can act in person or dispatch their own spokesperson to act as the chairman of the company, thus mastering the right to allocate resources of the company [27]. On the other hand, the controlling shareholders also have ability to exert influence on the management through the establishment of compensation mechanism or the appointment and removal of personnel [28]. In addition, the risk of control transfer will have an impact on the company's share price and performance. Once the controlling shareholder is replaced, the management may also be replaced. Therefore, the management also has the motivation to actively cooperate with the controlling shareholder to whitewash the company's performance by market value management and other means. With the prosperity of China's financial market, the return on investment in the financial industry is relatively high. The short-term boost of stock price pressure and the risk of control transfer have motivated the controlling shareholders of equity pledge to urge enterprises to allocate more financial assets. From the perspective of capital cost, enterprises will maximize the value by measuring the costs and benefits of investment projects when making investment decisions. According to the inherent characteristics of financial investment and real investment, financial asset investment has the advantages of short cycle and high liquidity, and enterprises can obtain excess profits from financial asset investment. However, the investment cycle of the real investment is long, it needs continuous and stable funds to support, and the return is uncertain. This leads to the cost of the enterprise investing in the financial asset project is far less than the cost of the investment in the real project. Therefore, the controlling shareholders tend to urge enterprises to increase their investment in financial assets during the period of equity pledge, as to stabilize the stock price and avoid the risk of control transfer.

Finally, the equity pledge of controlling shareholders will increase the financing costs of listed companies. There are two reasons. On the one hand, in order to avoid the transfer of control, the controlling shareholders will whitewash the company's operating data through market value management and other means [13]. The information asymmetry leads to investors' inability to accurately judge the company's future operating conditions, and then investors will require higher borrowing rates. On the other hand, based on the signal transmission theory, due to the concern about the agency problem and the risk of control transfer, external investors will regard the pledge of controlling shareholders' equity as a negative signal, so creditors will require a higher risk premium to compensate for their higher investment risk. When enterprises are faced with serious financing constraints, due to the need to pay a high cost to obtain external financing, enterprise investment is bound to be more sensitive to the company's cash flow [29], and the company will prefer financial assets with strong liquidity and short return period. Therefore, during the period of equity pledge of controlling shareholders, the company's financing costs will rise, which will aggravate the over-financing of enterprises.

Based on the above analysis, the following research assumptions are proposed:

H1: The equity pledge of controlling shareholders will aggravate the degree of excessive financialization of enterprises.

## Equity pledge, performance pressure and excessive financialization

Controlling shareholders' share pledge leads to the separation of control and cash flow rights, which means reducing the shared interests between major shareholders and minority shareholders. Thus it enhances the motivation of controlling shareholders to expropriate wealth from minority shareholders [24], which has a negative impact on the performance of listed companies [30], resulting in a large deviation between the actual operating performance of enterprises and expectations. The gap in business expectation strengthens the enterprise's risk motivation, resulting in short-sighted decision-making by the management [31, 32]. Considering that the technological progress of entities will not change significantly in the short term, and the return on investment of entities will not change abruptly, the short-sighted management may choose to allocate more high-yield financial assets to promote the improvement of corporate performance in the short term. Therefore, the performance pressure brought by the controlling shareholder's share pledge may aggravate the short-term behavior of the management, thus exacerbating the degree of excessive financialization of the enterprise.

Based on the above analysis, the following research assumptions are proposed:

H2: The equity pledge of controlling shareholders aggravates the degree of excessive financialization of enterprises through performance pressure.

## Research design

**Sample selection and data source.**   We take the A-share listed companies in China from 2010 to 2020 as the initial research sample, and screens the samples according to the following principles: ① Delete the samples of financial industry and ST companies; ② Delete the data that the shareholding ratio of major shareholders is less than 5% in the current year; ③ Delete the missing data of the sample; ④ Delete samples with only one year observation. In this paper, all continuous variables are shrunk by 1%, and the standard error of regression coefficient is clustered at the company level. Finally, 20902 company-annual observations were obtained. The data used in this paper are from CSMAR.

## Variable definition

**Explained variable: Degree of excessive financialization of enterprises.**   In order to measure the level of excessive financialization of enterprises, first of all, the ratio of financial assets to total assets is used to measure the degree of enterprise financialization. Financial assets include: trading financial assets, available-for-sale financial assets, investment real estate, derivative financial instruments, long-term equity investments, held-to-maturity investments, loans and advances. Then, refer to previous studies [33, 34], the following models are constructed to fit the normal financial level of physical enterprises:

$$Fin_{i,t} = \beta_0 + \beta_1 Fin_{i,t-1} + \beta_2 Growth_{i,t-1} + \beta_3 Lev_{i,t-1} + \beta_4 Cf_{i,t-1} + \beta_5 Size_{i,t-1}$$
$$+ \ \beta_6 Age_{i,t-1} + \beta_7 Roa_{i,t-1} + \sum Industry + \sum Year + \varepsilon_{i,t} \tag{1}$$

Here, $Fin_{i,t}$ indicates the degree of the entity's current financialization, $Fin_{i,t-1}$ indicates the degree of financialization of the entity in the previous period. $Growth_{i,t-1}$, $Lev_{i,t-1}$, $Cf_{i,t-1}$, $Size_{i,t-1}$, $Age_{i,t-1}$, $Roa_{i,t-1}$ represent the company's growth capacity, asset-liability ratio, free

cash flow, asset size, listing age and total return on assets in the year t-1. We also include industry and year fixed effects in the regression. Based on the above model (1), we use OLS regression to estimate the current degree of financialization of the enterprise, which is the normal level of financialization of the enterprise. The resulting residual is the degree of the deviation of the actual financial level of the enterprise from the normal financial level, which is the degree of over-financing denoted as *Exfin*. If *Exfin* is greater than 0, it means that the enterprise is over-financialized, and vice versa, it means that the enterprise is moderately financialized.

**Explanatory variable: Equity pledge of controlling shareholders.** According to the existing literature, this paper uses whether the controlling shareholders pledge their equity (*Pledge_d*) at the end of the year and the proportion of the controlling shareholders' equity pledge (*Pledge_r*) as the explanatory variables to measure the behavior and scale of the controlling shareholders' equity pledge.

**Control variables.** Referring to the existing literature, the following control variables are included in the model: the degree of the entity's financialization in the previous period ($Fin_{i,t-1}$), the level of the enterprise's financial leverage (*Lev*), the enterprise's cash flow (*Cf*), the enterprise's scale (*Size*), the enterprise's profitability (*Roa*), the proportion of major shareholders' shareholding (*Top*1), the asset turnover rate (*Assetturn*), the fixed asset ratio (*Fixed*), the market value (*TobinQ*), the combination of two functions (*Duality*), the executive compensation (*Pay*3), and the proportion of independent directors (*Indep*).

See Table 1 for the specific definitions of the main variables in this paper.

**Table 1. Definition of main variables.**

| Acronym | Variables | Explanation |
|---|---|---|
| *Exfin* | The degree of excessive financialization of enterprises | Actual financial level minus normal financial level |
| *Pledge_d* | Equity pledge of controlling shareholders | Dummy variable, if there is equity pledge of controlling shareholder at the end of the year, it is 1, otherwise it is 0 |
| *Pledge_r* | Share pledge ratio of controlling shareholders | The cumulative number of pledged shares that have not been decompressed by the controlling shareholders at the end of the year divided by the number of shares held by the controlling shareholders |
| $Fin_{i,t-1}$ | Actual financialization level of enterprises in the previous period | Non-monetary financial assets of the previous period divided by total assets of the previous period |
| *Lev* | Asset-liability ratio | Total liabilities at the end of the period divided by total assets at the end of the period |
| *Cf* | free cash flow | Net operating cash flow divided by total assets at the end of the period |
| *Size* | company size | The logarithm of the company's total assets |
| *Roa* | Return on total assets | Total profit divided by total assets |
| *Top*1 | Shareholding ratio of major shareholders | Proportion held by the largest shareholder in the annual report |
| *Assetturn* | Asset turnover rate | Operating income divided by total assets Closing balance |
| *Fixed* | Fixed asset ratio | Net fixed assets divided by total assets at the end of the period |
| *TobinQ* | Market value | Ratio of market value to total assets at the end of the year |
| *Duality* | Two functions in one | Dummy variable: 1 if the chairman concurrently serves as the general manager, otherwise 0 |
| *Pay*3 | Executive compensation | The logarithm of the total remuneration of the top three executives |
| *Indep* | Proportion of independent directors | Proportion of independent directors in the board of directors |

## Model settings

Before discussing the impact of controlling shareholders' equity pledge on the excessive financialization of enterprises, this paper first establishes a model (2) to empirically test how the controlling shareholders' equity pledge will affect the future main business performance of enterprises under the different circumstances of moderate and excessive financialization.

$$CorePerf_{i,t+1} = \beta_0 + \beta_1 Pledge\_d_{i,t} + \beta_2 Controls_{i,t} + \lambda_t + \mu_i + \varepsilon_{i,t} \tag{2}$$

Refer to the methods of Yong Du et al. [35] to measure the future main business performance of the enterprise. The specific calculation formula for the future main business performance of the enterprise is: (1) CorePerf1 = (operating profit—investment income—income from changes in fair value + investment income from associates and joint ventures) ÷ total assets; (2) CorePerf2 = (total profit—investment income—income from changes in fair value + investment income from associates and joint ventures) ÷ total assets from associates and joint ventures) ÷ total assets. Based on the degree of enterprise financialization, the whole sample is divided into excessive financialization sample and moderate financialization sample. According to model (2), the relationship between the equity pledge of controlling shareholders and the future main business performance of the enterprise is grouped and regressed.

Secondly, build a model (3) to test the impact of equity pledge of controlling shareholders on the level of enterprises' over-financing, where $i$ is the company and $t$ is the year. The explanatory variable $Pledge_{i,t}$ represents the equity pledge of the controlling shareholder, which is expressed by $Pledge\_d$ and $Pledge\_r$ respectively. $\mu_i$ and $\lambda_t$ are fixed effects and annual fixed effects at the company level. The two-way fixed effect model is used to control the inherent differences between the companies with and without equity pledge by controlling shareholders through the company fixed effect $\mu_i$, and eliminate the influence of factors that change with time through the annual fixed effect $\lambda_t$. The standard error estimate is adjusted by "clustering" at the company level.

$$Exfin_{i,t} = \beta_0 + \beta_1 Pledge_{i,t} + \beta_2 Controls_{i,t} + \lambda_t + \mu_i + \varepsilon_{i,t} \tag{3}$$

Finally, the model (4) is constructed to test the influence of equity pledge of controlling shareholders on whether the enterprise is over-financialized by using Logit regression as the robustness test of benchmark regression. Set the dummy variable *Exfin_dummy* to reflect whether the enterprise is over-financialized. When the actual level of financialization is greater than the normal level of financialization, it is over-financialized. At this time, *Exfin_dummy* value is 1, and vice versa, it is moderately financialized, and *Exfin_dummy* value is 0.

$$Exfin\_dummy_{i,t} = \beta_0 + \beta_1 Pledge_{i,t} + \beta_2 Controls_{i,t} + \sum Industry + \sum Year + \varepsilon_{i,t} \tag{4}$$

## Descriptive statistics

Table 2 shows the descriptive statistics of each variable. At the end of the year, the proportion of the observed value of equity pledge of controlling shareholders reached 42.8%, which indicates that more controlling shareholders have used equity pledge for financing. The standard deviation of the cumulative proportion of equity pledge during the year is 0.3334, indicating that the degree of equity pledge of different companies varies greatly. The maximum value of

**Table 2. Descriptive statistics of variables.**

| Variables | Mean | Median | St. Dev | Min | Max | Obs. |
|---|---|---|---|---|---|---|
| *Exfin* | -0.0003 | -0.0079 | 0.0516 | -0.4557 | 0.5197 | 20902 |
| *Pledge_d* | 0.4280 | 0.0000 | 0.4948 | 0.0000 | 1.0000 | 20902 |
| *Pledge_r* | 0.2359 | 0.0000 | 0.3334 | 0.0000 | 1.0000 | 20902 |
| $Fin_{i,t-1}$ | 0.0647 | 0.0268 | 0.0951 | 0.0000 | 0.4940 | 20902 |
| *Lev* | 0.4284 | 0.4237 | 0.1973 | 0.0594 | 0.8670 | 20902 |
| *Cf* | 0.0499 | 0.0479 | 0.0654 | -0.1350 | 0.2353 | 20902 |
| *Size* | 22.2371 | 22.0720 | 1.2291 | 20.0138 | 26.0536 | 20902 |
| *Roa* | 0.0455 | 0.0426 | 0.0657 | -0.2387 | 0.2318 | 20902 |
| *Top1* | 35.8580 | 34.1100 | 14.6323 | 9.3100 | 73.9600 | 20902 |
| *Assetturn* | 0.6418 | 0.5403 | 0.4285 | 0.0900 | 2.5677 | 20902 |
| *Fixed* | 0.2306 | 0.1983 | 0.1610 | 0.0050 | 0.7056 | 20902 |
| *TobinQ* | 2.0740 | 1.6513 | 1.3134 | 0.8621 | 8.4763 | 20902 |
| *Duality* | 0.2539 | 0.0000 | 0.4353 | 0.0000 | 1.0000 | 20902 |
| *Pay3* | 14.3727 | 14.3594 | 0.6799 | 12.7093 | 16.2094 | 20902 |
| *Indep* | 37.4198 | 33.3300 | 5.3030 | 33.3300 | 57.1400 | 20902 |

the excessive financialization index *Exfin* is 0.2002, the average value is -0.0004, and the standard deviation is 0.0443, which indicates that the actual financialization level of some sample companies exceeds the normal financialization level, and there are large differences between different enterprises' financialization levels.

## Empirical analysis

### Differences in the impact of moderate and excessive financialization on the economic consequences of equity pledge of controlling shareholders

Before discussing the impact of controlling shareholders' equity pledge on the excessive financialization of enterprises, this paper first examines whether there is any difference in the impact of controlling shareholders' equity pledge on the future main business performance of enterprises under the different circumstances of moderate and excessive financialization.

Based on the degree of enterprise financialization *Exfin*, the whole sample is divided into excessive financialization sample and moderate financialization sample according to the positive and negative signs of *Exfin*. Then, according to the model (2), the relationship between the equity pledge of the controlling shareholder and the future performance of the enterprise's main business is grouped and regressed, mainly focusing on the sign and significance of the estimated coefficient $\beta_1$ of *Pledge_d*. Table 3 reports the regression results of model (2). Columns (1)—(2) are the regression of the over-financialized group, and columns (3)—(4) are the regression of the moderately financialized group. The inhibitory effect of equity pledge of controlling shareholders on the future main business performance of enterprises only exists in the excessive financialization sample, while it is not significant in the moderate financialization sample. This is consistent with the views put forward in the introduction of this article: financial assets have the advantages of high income and high liquidity, and moderate financialization can promote the development of the main business of enterprises to a certain extent, while excessive financial asset investment will inhibit the development of the main business of enterprises due to crowding out the industrial investment of enterprises. Therefore, in the following research, we will focus on the excessive financialization behavior of enterprises.

**Table 3. The difference between moderate and excessive financialization.**

| Variables | (1) | (2) | (3) | (4) |
|---|---|---|---|---|
| | Excessive financialization | | Moderate financialization | |
| | *CorePerf*1 | *CorePerf*2 | *CorePerf*1 | *CorePerf*2 |
| *Pledge_d* | -0.00773** | -0.00870** | 0.00102 | 0.000784 |
| | (-2.0581) | (-2.3456) | (0.4375) | (0.3378) |
| $Fin_{i,t-1}$ | -0.0456** | -0.0437** | -0.0426*** | -0.0397** |
| | (-2.3537) | (-2.3055) | (-2.7029) | (-2.5140) |
| *Lev* | -0.000121 | 0.00328 | -0.00810 | -0.00495 |
| | (-0.0100) | (0.2762) | (-0.8007) | (-0.4936) |
| *Cf* | 0.0874*** | 0.0886*** | 0.0865*** | 0.0890*** |
| | (4.7868) | (4.8584) | (6.2166) | (6.4976) |
| *Size* | -0.0170*** | -0.0166*** | -0.0144*** | -0.0148*** |
| | (-4.6067) | (-4.3830) | (-5.3469) | (-5.4676) |
| *Roa* | 0.165*** | 0.156*** | 0.248*** | 0.244*** |
| | (5.8990) | (5.6120) | (9.4067) | (9.0597) |
| *Top*1 | 0.000347** | 0.000256* | 0.000607*** | 0.000619*** |
| | (2.2322) | (1.6820) | (4.9242) | (5.0598) |
| *Assetturn* | 0.0135** | 0.0124** | 0.00554 | 0.00292 |
| | (2.1861) | (1.9958) | (1.3355) | (0.7270) |
| *Fixed* | -0.00512 | -0.00364 | -0.0146 | -0.0163 |
| | (-0.3861) | (-0.2697) | (-1.3156) | (-1.4986) |
| *TobinQ* | 0.00559*** | 0.00547*** | 0.00474*** | 0.00496*** |
| | (4.1302) | (4.0168) | (5.0929) | (5.3000) |
| *Duality* | -0.00593 | -0.00504 | 0.000919 | 0.000574 |
| | (-1.5234) | (-1.2557) | (0.3251) | (0.2041) |
| *Pay3* | 0.00720** | 0.00791** | 0.00396* | 0.00367 |
| | (2.2529) | (2.4541) | (1.6733) | (1.5753) |
| *Indep* | 0.0000488 | 0.000131 | 0.000516** | 0.000499** |
| | (0.2013) | (0.5090) | (2.5252) | (2.4432) |
| *Constant* | 0.261*** | 0.251*** | 0.235*** | 0.256*** |
| | (2.9161) | (2.7431) | (3.7885) | (4.1079) |
| Year fixed effect | YES | YES | YES | YES |
| Firm fixed effect | YES | YES | YES | YES |
| N | 5351 | 5351 | 10377 | 10377 |
| R2_a | 0.135 | 0.139 | 0.115 | 0.120 |

Note: This table shows the results of testing the influence of equity pledge of controlling shareholders on the future main business performance of enterprises under different situations of excessive financialization and moderate financialization. The definitions of the variables are provided in Table 1. Firm and year fixed effects are included in the models. The coefficient estimates and t-statistics are reported based on robust standard errors clustered by firm.

*** $p < 0.01$,

** $p < 0.05$,

* $p < 0.1$; in parentheses are the t-statistics of the corresponding coefficients.

## Basic regression analysis

Table 4 reports the regression results of the equity pledge of controlling shareholders and the excessive financialization of enterprises. The explained variable in columns (1) and (2) is the degree of over-financing *Exfin*. According to the setting of model (3), the two-way fixed effect model is used for regression, which is also the main regression of this paper. The explained

**Table 4. Controlling shareholder's equity pledge and enterprise's over-financing.**

| Variables | (1) | (2) | (3) | (4) |
|---|---|---|---|---|
| | Exfin | Exfin | Exfin_dummy | Exfin_dummy |
| Pledge_d | 0.0041*** | | 0.1020*** | |
| | (3.0026) | | (3.0582) | |
| Pledge_r | | 0.0069*** | | 0.1070** |
| | | (3.3265) | | (2.1965) |
| $Fin_{i,t-1}$ | -0.3485*** | -0.3486*** | 5.2152*** | 5.2025*** |
| | (-23.5546) | (-23.5227) | (27.3383) | (27.2950) |
| Lev | -0.0019 | -0.0021 | 0.7594*** | 0.7643*** |
| | (-0.3153) | (-0.3436) | (6.9405) | (6.9717) |
| Cf | 0.0269*** | 0.0269*** | 1.7976*** | 1.7996*** |
| | (3.3438) | (3.3362) | (6.3243) | (6.3338) |
| Size | -0.0064*** | -0.0063*** | 0.0287 | 0.0279 |
| | (-3.8255) | (-3.7762) | (1.4749) | (1.4295) |
| Roa | -0.0366*** | -0.0363*** | -1.3599*** | -1.3379*** |
| | (-3.6354) | (-3.6031) | (-4.5194) | (-4.4497) |
| Top1 | -0.0001 | -0.0001 | -0.0004 | -0.0003 |
| | (-1.0613) | (-1.0381) | (-0.3398) | (-0.2926) |
| Assetturn | -0.0061** | -0.0061** | -0.0940** | -0.0976** |
| | (-2.3071) | (-2.3004) | (-2.0604) | (-2.1396) |
| Fixed | -0.0524*** | -0.0521*** | -0.9723*** | -0.9711*** |
| | (-7.1635) | (-7.1332) | (-7.4030) | (-7.3979) |
| TobinQ | 0.0015** | 0.0016** | 0.0265* | 0.0264* |
| | (2.3956) | (2.4517) | (1.7519) | (1.7469) |
| Duality | 0.0011 | 0.0011 | 0.0275 | 0.0327 |
| | (0.6614) | (0.6648) | (0.7350) | (0.8762) |
| Pay3 | -0.0003 | -0.0003 | 0.0083 | 0.0067 |
| | (-0.2328) | (-0.2386) | (0.2897) | (0.2311) |
| Indep | -0.0001 | -0.0001 | 0.0006 | 0.0007 |
| | (-1.0997) | (-1.0967) | (0.1938) | (0.2349) |
| Constant | 0.1830*** | 0.1810*** | -2.1271*** | -2.0877*** |
| | (4.3912) | (4.3498) | (-4.3289) | (-4.2449) |
| Year fixed effect | YES | YES | YES | YES |
| Firm fixed effect | YES | YES | | |
| Industry fixed effect | | | YES | YES |
| N | 20902 | 20902 | 20902 | 20902 |
| R2 | 0.1321 | 0.1323 | 0.0596 | 0.0594 |

Note: This table shows the results of testing the influence of equity pledge of controlling shareholders on excessive financialization of enterprises. The definitions of the variables are provided in Table 1. Firm and year fixed effects are included in the models. The coefficient estimates and t-statistics are reported based on robust standard errors clustered by firm.

*** $p < 0.01$,

** $p < 0.05$,

* $p < 0.1$; in parentheses are the t-statistics of the corresponding coefficients.

variable in columns (3) and (4) is the over-financialized dummy variable *Exfin_dummy*. According to the setting of model (4), the Logit model is used for regression as the robustness test of this paper. The data shows that whether the controlling shareholders pledge their equity (*Pledge_d*) and the proportion of the pledge of the controlling shareholders (*Pledge_r*) are

significantly and positively correlated with the company's over-financialization indicators *Exfin* and *Exfin_dummy*, indicating that after the shareholding pledge of the controlling shareholders, the company excessively uses the funds to allocate financial assets. The main assumption H1 of this paper is verified.

## Impact mechanism test

In the part of theoretical analysis, we believe that the actual performance of the enterprise after the equity pledge of the controlling shareholder is lower than expected, and the performance pressure caused by the expected gap has triggered the short-sighted behavior of the management, which makes the management willing to conspire with the controlling shareholder to cause the excessive financialization of the enterprise. If this logic is true, it can be expected that during the period of controlling shareholder pledging, with the expansion of the company's performance expectation gap, the degree of excessive financialization of companies with serious management short-sighted behavior will be greater. Since management myopia is difficult to be directly measured, this paper tests the samples by grouping according to the influencing factors of management myopia, and groups them respectively from two aspects: management shareholding ratio and institutional investor shareholding ratio. Among them, a large proportion of the management's shareholding means that the interests of the management and the company are consistent, so the management gives priority to the long-term interests of the enterprise when making decisions, thus reducing the short-term behavior. At the same time, most institutional investors in China pursue short-term interests, which means that institutional investors' shareholding will bring certain performance pressure to the company's management and drive their short-sighted investment behavior [15]. Therefore, this paper starts from two aspects to test. ① We test whether the equity pledge of controlling shareholders has increased the gap between the actual performance and the expected performance of the company. ② From the perspective of management's shareholding level and institutional investor's shareholding ratio, we test whether the companies with more short-sighted behaviors are equipped with more financial assets than those with less serious management's short-sighted behaviors in the case of controlling shareholder's equity pledge. If the equity pledge of controlling shareholders increases the gap of performance expectations, then when the company holds less shares for the management and higher shares for institutional investors, the management would be more sensitive to the increase of the gap of performance expectations of the company. This will result in the greater impact of the equity pledge of controlling shareholders on the degree of enterprise financialization.

Measure the performance gap of enterprises *Hps* by referring to the methods of Chen et al. [36]. If the actual performance of the current year is lower than the historical expected performance of the current year, then *Hps* is the absolute value of the gap between the two; otherwise, *Hps* is 0. The management shareholding ratio and the institutional investor shareholding ratio are used to measure the management shareholding level (*Mngmhldn*) and the institutional investor shareholding level (*InsInvestorProp*) respectively. Divide the sample into high and low groups according to the median management shareholding ratio. Divide the sample into five groups according to the shareholding ratio of institutional investors. Select the two groups with the highest and lowest shareholding ratio of institutional investors for grouping test.

First of all, this paper examines the impact of equity pledge of controlling shareholders on the gap of corporate performance expectations. The (1) and (2) columns in Table 5 are the regression results. The equity pledge of the controlling shareholder has increased the gap of the enterprise's performance expectations and increased the performance pressure of the

**Table 5. Mechanism test.**

| Variables | Full sample | | Low shareholding level of management | | High shareholding level of management | |
|---|---|---|---|---|---|---|
| | (1) | (2) | (3) | (4) | (5) | (6) |
| | *Hps* | *Hps* | *Exfin* | *Exfin* | *Exfin* | *Exfin* |
| *Pledge_d* | 0.00914** | | 0.0059*** | | 0.0018 | |
| | (1.9950) | | (2.6535) | | (0.9679) | |
| *Pledge_r* | | 0.0117* | | 0.0097*** | | 0.0033 |
| | | (1.6504) | | (2.6291) | | (1.2241) |
| *Fin$_{i,t-1}$* | -0.00438 | -0.00382 | -0.3497*** | -0.3492*** | -0.4400*** | -0.4403*** |
| | (-0.5136) | (-0.4489) | (-14.9945) | (-14.9639) | (-19.7759) | (-19.7572) |
| *Lev* | -0.0430*** | -0.0430*** | -0.0178** | -0.0179** | 0.0036 | 0.0035 |
| | (-6.3721) | (-6.5023) | (-2.0578) | (-2.0704) | (0.4024) | (0.3933) |
| *Cf* | 0.0210* | 0.0211* | 0.0198* | 0.0200* | 0.0307** | 0.0306** |
| | (1.6537) | (1.6511) | (1.8158) | (1.8338) | (2.5361) | (2.5246) |
| *Size* | -0.00188 | -0.00152 | -0.0070*** | -0.0069*** | -0.0061** | -0.0061** |
| | (-0.4621) | (-0.3838) | (-2.7569) | (-2.7285) | (-2.2979) | (-2.3002) |
| *Roa* | -0.694*** | -0.693*** | -0.0281* | -0.0284** | -0.0351** | -0.0348** |
| | (-29.0948) | (-29.0187) | (-1.9451) | (-1.9651) | (-2.4257) | (-2.4020) |
| *Top1* | -0.0000168 | -0.0000134 | -0.0001 | -0.0001 | 0.0001 | 0.0001 |
| | (-0.2298) | (-0.1833) | (-0.9715) | (-0.9569) | (0.5512) | (0.5560) |
| *Assetturn* | 0.0121** | 0.0122** | -0.0048 | -0.0048 | -0.0057 | -0.0057 |
| | (2.5406) | (2.5505) | (-1.1639) | (-1.1621) | (-1.4421) | (-1.4326) |
| *Fixed* | -0.0376** | -0.0370** | -0.0523*** | -0.0525*** | -0.0444*** | -0.0441*** |
| | (-2.1227) | (-2.1018) | (-5.4552) | (-5.4706) | (-3.9077) | (-3.8803) |
| *TobinQ* | 0.00542*** | 0.00551*** | 0.0003 | 0.0003 | 0.0017* | 0.0017* |
| | (4.8986) | (4.9532) | (0.3868) | (0.3686) | (1.7960) | (1.8365) |
| *Duality* | -0.00334 | -0.00334 | 0.0009 | 0.0009 | -0.0010 | -0.0010 |
| | (-1.0059) | (-1.0060) | (0.3708) | (0.3783) | (-0.4809) | (-0.4764) |
| *Pay3* | 0.0117*** | 0.0117*** | -0.0006 | -0.0007 | 0.0017 | 0.0017 |
| | (5.8642) | (5.8790) | (-0.3523) | (-0.3802) | (0.7101) | (0.7136) |
| *Indep* | -0.0000939 | -0.0000911 | -0.0002* | -0.0002* | 0.0001 | 0.0001 |
| | (-0.6542) | (-0.6375) | (-1.6767) | (-1.6551) | (0.6778) | (0.6731) |
| *Constant* | -0.0522 | -0.0594 | 0.2211*** | 0.2199*** | 0.1202* | 0.1194* |
| | (-0.6157) | (-0.7129) | (3.7611) | (3.7495) | (1.8988) | (1.8972) |
| Year / Firm fixed effect | YES | YES | YES | YES | YES | YES |
| N | 20902 | 20902 | 10592 | 10592 | 10310 | 10310 |
| R2_a | 0.156 | 0.156 | 0.1378 | 0.1381 | 0.1802 | 0.1803 |

Note: This table shows the test results of the mechanism test. The definitions of the variables are provided in Table 1. Firm and year fixed effects are included in the models. The coefficient estimates and t-statistics are reported based on robust standard errors clustered by firm.

*** p < 0.01,

** p < 0.05,

* p < 0.1; in parentheses are the t-statistics of the corresponding coefficients.

management. Further, this paper conducts group regression according to the shareholding ratio of management and institutional investors, and the results are shown in columns (3)–(6) of Tables 5 and 6. In the case of low shareholding ratio of management and high shareholding ratio of institutional investors, equity pledge of controlling shareholders has a stronger impact

**Table 6. Mechanism test.**

| Variables | Low shareholding ratio of institutional investors | | High shareholding ratio of institutional investors | |
|---|---|---|---|---|
| | (1) | (2) | (3) | (4) |
| | *Exfin* | *Exfin* | *Exfin* | *Exfin* |
| *Pledge_d* | 0.0008 | | 0.0090*** | |
| | (0.2739) | | (2.7873) | |
| *Pledge_r* | | -0.0021 | | 0.0210*** |
| | | (-0.4433) | | (3.0896) |
| $Fin_{i,t-1}$ | -0.4755*** | -0.4756*** | -0.4236*** | -0.4234*** |
| | (-13.9341) | (-13.9477) | (-11.4332) | (-11.5817) |
| *Lev* | -0.0054 | -0.0050 | 0.0058 | 0.0051 |
| | (-0.3763) | (-0.3513) | (0.4161) | (0.3684) |
| *Cf* | 0.0268 | 0.0270 | 0.0160 | 0.0140 |
| | (1.4529) | (1.4608) | (0.9012) | (0.7896) |
| *Size* | -0.0089* | -0.0087* | -0.0049 | -0.0048 |
| | (-1.7754) | (-1.7452) | (-0.9645) | (-0.9725) |
| *Roa* | -0.0179 | -0.0180 | -0.0332 | -0.0307 |
| | (-0.8722) | (-0.8742) | (-1.0093) | (-0.9305) |
| *Top1* | 0.0003 | 0.0004* | 0.0002 | 0.0002 |
| | (1.6280) | (1.6696) | (0.8075) | (0.7292) |
| *Assetturn* | -0.0242*** | -0.0245*** | 0.0051 | 0.0044 |
| | (-3.6000) | (-3.6166) | (0.8730) | (0.7586) |
| *Fixed* | -0.0443** | -0.0436** | -0.0710*** | -0.0696*** |
| | (-2.3378) | (-2.3060) | (-3.9690) | (-3.9663) |
| *TobinQ* | 0.0022 | 0.0022 | 0.0011 | 0.0011 |
| | (1.0577) | (1.0598) | (0.7952) | (0.7932) |
| *Duality* | 0.0018 | 0.0018 | 0.0035 | 0.0033 |
| | (0.5829) | (0.5880) | (0.7161) | (0.6930) |
| *Pay3* | -0.0021 | -0.0021 | -0.0005 | -0.0009 |
| | (-0.4905) | (-0.4876) | (-0.1549) | (-0.2965) |
| *Indep* | -0.0003 | -0.0002 | -0.0005* | -0.0005* |
| | (-0.8700) | (-0.8497) | (-1.8162) | (-1.7989) |
| *Constant* | 0.2283* | 0.2233* | 0.1529 | 0.1591 |
| | (1.8862) | (1.8499) | (1.4572) | (1.5380) |
| Year fixed effect | YES | YES | YES | YES |
| Firm fixed effect | YES | YES | YES | YES |
| N | 4177 | 4177 | 4175 | 4175 |
| R2_a | 0.1950 | 0.1950 | 0.1871 | 0.1896 |

Note: This table shows the test results of the mechanism test. The definitions of the variables are provided in Table 1. Firm and year fixed effects are included in the models. The coefficient estimates and t-statistics are reported based on robust standard errors clustered by firm.

*** $p < 0.01$,

** $p < 0.05$,

* $p < 0.1$; in parentheses are the t-statistics of the corresponding coefficients.

on the over-financing of enterprises. The above results support the performance pressure mechanism, that is, the actual performance of the company is lower than expected due to the equity pledge of the controlling shareholder, which increases the performance pressure of the management and urges the company to invest more financial assets.

## Robustness test

### Endogenous test

In order to solve the endogenous problem caused by potential missing variables, use two-stage regression method (2SLS). Select the average pledge level of the industry in the same year (*indPledge_d* and *indPledge_r*) and the average pledge level of the province in the same year (*ProPledge_d* and *ProPledge_r*) as the instrumental variables of the degree of equity pledge of the controlling shareholder (*Pledge_d* and *Pledge_r*). Table 7 shows the results of the two stages of regression, and (1) and (3) show the results of the first stage of regression. The average pledge level of the industry and province is positively correlated with the pledge level of the company at the level of 1%. The regression results of the second stage are listed in (2) and (4). After overcoming the endogenous problem, the coefficients of *Pledge_d* and *Pledge_r* are still significantly positive. The above results show that after overcoming the endogenous problems, the equity pledge of controlling shareholders still has a significant role in promoting the over-financing of enterprises, and the previous conclusion is sound.

### Other robustness tests

In addition, in order to ensure the reliability of the main research conclusions, this paper also carries out robustness tests from the following two aspects. First, the core explanatory variable lags for one period. Considering that the influence of equity pledge of controlling shareholders on the degree of over-financialization of enterprises may have a lag effect, and in order to alleviate the endogenous problem caused by reverse causality, this paper lags the core explanatory variable (*Pledge_d* and *Pledge_r*) of basic regression analysis by one period, and use model (2) to carry out regression again. The results are shown in columns (1) and (2) of Table 8. The equity pledge of controlling shareholders is significantly and positively correlated with the degree of over-financing of enterprises, which is consistent with the previous conclusion. Second, change the explanatory variables. The previous article uses the ratio of the number of equity pledge of the controlling shareholders at the end of the year to the number of shares held by the controlling shareholders (*Pledge_r*) as the explanatory variable for analysis. This part uses the percentage of the number of equity pledge of the controlling shareholders at the end of the year to the total shares of the company (*Pledge_r*2) to replace the *Pledge_r* used above for robustness test. The regression results are shown in column (3) of Table 8, which shows that the more the number of equity pledge of controlling shareholders, the higher the degree of over-financing of enterprises, and the main regression results of this paper remain unchanged.

## Further analysis

### The influence of economic policy uncertainty on the relationship between controlling shareholder's equity pledge and enterprise's excessive financialization

When enterprises are in an environment with high economic policy uncertainty, investment decisions will be affected to some extent. First of all, the uncertainty of economic policy will reduce the probability of success of enterprises' projects, and make the rate of return of enterprises' investment in fixed assets decline, which will inhibit the investment in fixed assets [37]. Compared with traditional physical investment, financial asset investment has strong liquidity and low conversion cost. When enterprises are in short of funds, they can quickly sell financial assets to obtain cash flow. The existing literature shows that when the uncertainty of economic policy is high, the market demand is more difficult to predict, and enterprises will increase their

**Table 7. 2SLS regression results.**

| Variables | One stage | Two stages | One stage | Two stages |
|---|---|---|---|---|
| | **(1)** | **(2)** | **(3)** | **(4)** |
| | *Pledge_d* | *Fin* | *Pledge_r* | *Fin* |
| Pledge_d | | 0.014* | | |
| | | (1.87) | | |
| indPledge_d | 0.558*** | | | |
| | (14.72) | | | |
| ProPledge_d | 0.564*** | | | |
| | (10.00) | | | |
| Pledge_r | | | | 0.017* |
| | | | | (1.65) |
| indPledge_r | | | 0.582*** | |
| | | | (13.99) | |
| ProPledge_r | | | 0.555*** | |
| | | | (9.32) | |
| $Fin_{i,t-1}$ | 0.262*** | -0.351*** | 0.170*** | -0.351*** |
| | (4.21) | (-23.64) | (3.92) | (-23.63) |
| Lev | 0.098** | -0.003 | 0.085*** | -0.003 |
| | (2.30) | (-0.51) | (2.88) | (-0.51) |
| Cf | 0.030 | 0.026*** | 0.028 | 0.026*** |
| | (0.61) | (3.24) | (0.81) | (3.25) |
| Size | 0.081*** | -0.007*** | 0.038*** | -0.007*** |
| | (6.52) | (-3.88) | (4.19) | (-3.81) |
| Roa | -0.027 | -0.036*** | -0.067 | -0.036*** |
| | (-0.40) | (-3.60) | (-1.40) | (-3.53) |
| Top1 | 0.000 | -0.000 | -0.000 | -0.000 |
| | (0.56) | (-1.04) | (-0.33) | (-0.99) |
| Assetturn | 0.008 | -0.006** | 0.004 | -0.006** |
| | (0.35) | (-2.30) | (0.24) | (-2.29) |
| Fixed | 0.019 | -0.053*** | -0.041 | -0.052*** |
| | (0.36) | (-7.16) | (-1.02) | (-7.08) |
| TobinQ | 0.014*** | 0.001** | 0.006** | 0.002** |
| | (3.57) | (2.13) | (2.14) | (2.37) |
| Duality | -0.001 | 0.001 | -0.000 | 0.001 |
| | (-0.07) | (0.67) | (-0.01) | (0.68) |
| Pay3 | 0.002 | -0.000 | 0.004 | -0.000 |
| | (0.19) | (-0.25) | (0.50) | (-0.26) |
| Indep | 0.001 | -0.000 | 0.000 | -0.000 |
| | (0.75) | (-1.18) | (0.50) | (-1.15) |
| Constant | -1.878*** | 0.2173*** | -0.914*** | 0.2059*** |
| | (-6.19) | (7.37) | (-4.26) | (7.33) |
| Year fixed effect | YES | YES | YES | YES |
| Firm fixed effect | YES | YES | YES | YES |
| N | 20,902 | 20,902 | 20,902 | 20,902 |
| Adj-R2 | 0.092 | 0.130 | 0.090 | 0.131 |

Note: This table shows the results of endogeneity test. The definitions of the variables are provided in Table 1. Firm and year fixed effects are included in the models. The coefficient estimates and t-statistics are reported based on robust standard errors clustered by firm.

*** $p < 0.01$,

** $p < 0.05$,

* $p < 0.1$; in parentheses are the t-statistics of the corresponding coefficients.

**Table 8. Other robustness tests.**

| Variables | (1) | (2) | (3) |
|---|---|---|---|
|  | *Exfin* | *Exfin* | *Exfin* |
| *Pledge_d_1* | 0.00582*** |  |  |
|  | (3.5129) |  |  |
| *Pledge_r_1* |  | 0.00733*** |  |
|  |  | (2.8079) |  |
| *Pledge_r2* |  |  | 0.0238*** |
|  |  |  | (3.9863) |
| $Fin_{i,t-1}$ | -0.339*** | -0.339*** | -0.3485*** |
|  | (-20.3238) | (-20.2861) | (-23.5106) |
| *Lev* | -0.00437 | -0.00464 | -0.0023 |
|  | (-0.6203) | (-0.6566) | (-0.3735) |
| *Cf* | 0.0308*** | 0.0306*** | 0.0271*** |
|  | (3.3687) | (3.3539) | (3.3579) |
| *Size* | -0.00846*** | -0.00825*** | -0.0062*** |
|  | (-4.2447) | (-4.1492) | (-3.7499) |
| *Roa* | -0.0306*** | -0.0306*** | -0.0367*** |
|  | (-2.7624) | (-2.7600) | (-3.6481) |
| *Top1* | -0.0000439 | -0.0000453 | -0.0001 |
|  | (-0.3823) | (-0.3939) | (-1.4383) |
| *Assetturn* | -0.00876*** | -0.00876*** | -0.0061** |
|  | (-3.0445) | (-3.0403) | (-2.3032) |
| *Fixed* | -0.0558*** | -0.0554*** | -0.0523*** |
|  | (-6.4182) | (-6.3853) | (-7.1618) |
| *TobinQ* | 0.00111 | 0.00115 | 0.0016** |
|  | (1.4015) | (1.4417) | (2.4243) |
| *Duality* | 0.00157 | 0.00154 | 0.0011 |
|  | (0.8874) | (0.8698) | (0.6806) |
| *Pay3* | -0.000518 | -0.000493 | -0.0003 |
|  | (-0.3298) | (-0.3135) | (-0.2076) |
| *Indep* | -0.000188 | -0.000190 | -0.0001 |
|  | (-1.4650) | (-1.4764) | (-1.1639) |
| *Constant* | 0.238*** | 0.233*** | 0.1807*** |
|  | (4.8555) | (4.7713) | (4.3483) |
| Year fixed effect | YES | YES | YES |
| Firm fixed effect | YES | YES | YES |
| N | 17045 | 17045 | 20902 |
| R2_a | 0.126 | 0.126 | 0.1325 |

Note: This table shows the robustness test results. The definitions of the variables are provided in Table 1. Firm and year fixed effects are included in the models. The coefficient estimates and t-statistics are reported based on robust standard errors clustered by firm.

*** $p < 0.01$,

** $p < 0.05$,

* $p < 0.1$; in parentheses are the t-statistics of the corresponding coefficients.

holdings of financial assets for the purpose of preventive savings [18]. Therefore, when the uncertainty of economic policy is high, the controlling shareholders who pledge their equity may be inclined to guide enterprises to hold more financial assets to cope with the impact of the external environment on the entity business and reduce the risk of control transfer.

However, the rising uncertainty of economic policy may also inhibit the excessive allocation of financial assets by enterprises with equity pledge of controlling shareholders. The research of Yuchao Peng, Xun Han and Jianjun Li [19] shows that the uncertainty of economic policy has significantly inhibited the trend of enterprise financialization. This is because in China, enterprises increase their investment in financial assets not because of the precautionary saving motive, but to pursue high profits in the financial market. The increase of economic policy uncertainty has exacerbated the volatility of the financial market. In order to avoid risks, the controlling shareholders who pledge their equity may reduce their financial investment.

Based on the above analysis, this paper refers to the monthly China's economic policy uncertainty index constructed by Baker et al. [38] to measure economic policy uncertainty. The uncertainty of economic policy is divided into high and low groups according to the median for grouping regression. Table 9 reports the regression results. The group with low uncertainty of economic policy is significantly positive, while the group with high uncertainty is negative and not significant. The above results show that the lower the uncertainty of economic policy or the relatively stable economic policy, the greater the positive impact of equity pledge of controlling shareholders on the over-financing of enterprises.

## The impact of product market competition on the relationship between controlling shareholders' equity pledge and corporate over-financialization

The level of product market competition will affect the investment strategy of enterprises. With the intensification of product market competition, the company is bound to increase research and development investment to achieve product differentiation or reduce costs, as to gain and maintain competitive advantage [39]. In addition, the fierce product market competition can bring the threat of bankruptcy liquidation, which will play a certain role in inhibiting the short-term opportunistic behavior of the management [40]. The stronger the competition in the industry where the enterprise is located, in order to improve the product competitiveness and prevent the risk of control transfer caused by the decline of share price, the controlling shareholders who pledge the equity will make reasonable expectations for the long-term development of the enterprise and use more resources in the physical investment. In a weak competitive environment, due to the monopoly position of enterprises in the industry, the management and shareholders of equity pledge enterprises may be more short-sighted, and spend more resources on financial investment to cope with the downward pressure of share prices caused by equity pledge. Therefore, the fierce competition in the product market may alleviate the positive relationship between the equity pledge of controlling shareholders and the enterprise's financialization.

The above is the analysis of the investment strategy selected by enterprises. Now consider the financing risk of enterprises. The fierce competition will increase the fluctuation of operating cash flow [41], and the pledge of controlling shareholders' equity will lead to higher financing costs [9]. At this time, holding more cash will help the company resist risks [42, 43]. Because financial assets have the characteristics of strong liquidity and low conversion cost, they can replace cash holding to some extent. As a result, the controlling shareholder equity pledge enterprises are in an increasingly competitive industry. In order to reduce the risk of cash flow shortage, they may hold more financial assets. Therefore, the competition in the product market may aggravate the positive relationship between the equity pledge of controlling shareholders and the enterprise's financialization.

The Herfindahl index calculated by operating income is used to measure the product market competition, and the sample is divided into high and low groups for regression analysis. Table 10 shows that in the less competitive group, the regression coefficient of *Pledge_d* and

**Table 9. Impact of economic policy uncertainty.**

| Variables | Low uncertainty | | High uncertainty | |
|---|---|---|---|---|
| | (1) | (2) | (3) | (4) |
| | *Exfin* | *Exfin* | *Exfin* | *Exfin* |
| *Pledge_d* | 0.0078*** | | -0.0024 | |
| | (3.8976) | | (-1.0340) | |
| *Pledge_r* | | 0.0141*** | | -0.0015 |
| | | (4.0965) | | (-0.4809) |
| $Fin_{i,t-1}$ | -0.4613*** | -0.4605*** | -0.5624*** | -0.5625*** |
| | (-18.1127) | (-18.1449) | (-28.5630) | (-28.5668) |
| *Lev* | 0.0006 | 0.0003 | -0.0207* | -0.0206* |
| | (0.0660) | (0.0294) | (-1.8176) | (-1.8118) |
| *Cf* | 0.0105 | 0.0098 | 0.0390*** | 0.0389*** |
| | (1.1934) | (1.1166) | (2.7207) | (2.7185) |
| *Size* | -0.0134*** | -0.0131*** | -0.0084** | -0.0085** |
| | (-4.6205) | (-4.5631) | (-2.1247) | (-2.1586) |
| *Roa* | -0.0455*** | -0.0446*** | -0.0004 | -0.0004 |
| | (-3.0112) | (-2.9624) | (-0.0316) | (-0.0344) |
| *Top1* | -0.0001 | -0.0001 | -0.0000 | -0.0000 |
| | (-0.7656) | (-0.7703) | (-0.0972) | (-0.1132) |
| *Assetturn* | 0.0029 | 0.0031 | -0.0069* | -0.0069* |
| | (0.9300) | (0.9863) | (-1.8557) | (-1.8520) |
| *Fixed* | -0.0351*** | -0.0345*** | -0.0817*** | -0.0817*** |
| | (-4.0213) | (-3.9688) | (-5.2140) | (-5.2091) |
| *TobinQ* | 0.0003 | 0.0003 | 0.0032*** | 0.0032*** |
| | (0.5271) | (0.5523) | (2.8278) | (2.8196) |
| *Duality* | -0.0026 | -0.0025 | 0.0010 | 0.0011 |
| | (-1.1387) | (-1.0894) | (0.4031) | (0.4070) |
| *Pay3* | 0.0001 | -0.0001 | 0.0021 | 0.0021 |
| | (0.0404) | (-0.0491) | (0.8981) | (0.9000) |
| *Indep* | 0.0002 | 0.0002 | -0.0002 | -0.0002 |
| | (1.3437) | (1.3519) | (-0.8242) | (-0.8343) |
| *Constant* | 0.3167*** | 0.3125*** | 0.2162** | 0.2184** |
| | (4.6818) | (4.6715) | (2.3865) | (2.4127) |
| Year fixed effect | YES | YES | YES | YES |
| Firm fixed effect | YES | YES | YES | YES |
| N | 9050 | 9050 | 11852 | 11852 |
| R2_a | 0.2026 | 0.2035 | 0.2567 | 0.2567 |

Note: This table shows the results of the tests grouped by economic policy uncertainty. The definitions of the variables are provided in Table 1. Firm and year fixed effects are included in the models. The coefficient estimates and t-statistics are reported based on robust standard errors clustered by firm.

*** $p < 0.01$,

** $p < 0.05$,

* $p < 0.1$; in parentheses are the t-statistics of the corresponding coefficients.

*Pledge_r* are significantly positive at the level of 1%, while in the more competitive group, the regression coefficient of *Pledge_d* is positive but not significant, and the regression coefficient of *Pledge_r* is only positive at the level of 10%. The above results show that the lower the competition intensity of the product market, the greater the positive impact of the

**Table 10. Impact of product market competition.**

| Variables | Less competitive | | Fierce competition | |
|---|---|---|---|---|
| | (1) | (2) | (3) | (4) |
| | *Exfin* | *Exfin* | *Exfin* | *Exfin* |
| *Pledge_d* | 0.0077*** | | 0.0013 | |
| | (3.6671) | | (0.6548) | |
| *Pledge_r* | | 0.0105*** | | 0.0056* |
| | | (3.3939) | | (1.8015) |
| $Fin_{i,t-1}$ | -0.3528*** | -0.3518*** | -0.3468*** | -0.3476*** |
| | (-17.7527) | (-17.7118) | (-15.2404) | (-15.2736) |
| *Lev* | -0.0105 | -0.0110 | 0.0083 | 0.0082 |
| | (-1.1279) | (-1.1701) | (1.0101) | (0.9992) |
| *Cf* | 0.0196* | 0.0195* | 0.0300** | 0.0303** |
| | (1.7743) | (1.7581) | (2.5331) | (2.5581) |
| *Size* | -0.0060** | -0.0056** | -0.0062** | -0.0065*** |
| | (-2.5440) | (-2.3787) | (-2.5080) | (-2.6134) |
| *Roa* | -0.0343** | -0.0340** | -0.0296** | -0.0292** |
| | (-2.1288) | (-2.1051) | (-2.1805) | (-2.1617) |
| *Top1* | -0.0001 | -0.0001 | -0.0002 | -0.0002 |
| | (-0.5641) | (-0.5445) | (-1.1350) | (-1.1247) |
| *Assetturn* | -0.0066* | -0.0065* | -0.0087* | -0.0089* |
| | (-1.8710) | (-1.8561) | (-1.7487) | (-1.7881) |
| *Fixed* | -0.0572*** | -0.0564*** | -0.0391*** | -0.0391*** |
| | (-5.2138) | (-5.1606) | (-3.9141) | (-3.8990) |
| *TobinQ* | 0.0041*** | 0.0041*** | -0.0003 | -0.0003 |
| | (3.9544) | (3.9730) | (-0.3607) | (-0.3435) |
| *Duality* | 0.0017 | 0.0017 | 0.0013 | 0.0013 |
| | (0.6815) | (0.6907) | (0.6259) | (0.6150) |
| *Pay3* | -0.0019 | -0.0019 | 0.0016 | 0.0016 |
| | (-0.8780) | (-0.8612) | (0.7880) | (0.7976) |
| *Indep* | -0.0002 | -0.0002 | 0.0001 | 0.0001 |
| | (-1.3080) | (-1.2894) | (0.3315) | (0.3365) |
| *Constant* | 0.1986*** | 0.1902*** | 0.1469** | 0.1511*** |
| | (3.2882) | (3.1493) | (2.5523) | (2.6344) |
| Year fixed effect | YES | YES | YES | YES |
| Firm fixed effect | YES | YES | YES | YES |
| N | 10441 | 10441 | 10461 | 10461 |
| R2_a | 0.1449 | 0.1447 | 0.1201 | 0.1206 |

Note: This table shows the results of tests grouped by product market competition. The definitions of the variables are provided in Table 1. Firm and year fixed effects are included in the models. The coefficient estimates and t-statistics are reported based on robust standard errors clustered by firm.

*** $p < 0.01$,

** $p < 0.05$,

* $p < 0.1$; in parentheses are the t-statistics of the corresponding coefficients.

shareholding pledge of the controlling shareholders on the over-financing of enterprises, that is, the fierce competition in the product market can alleviate the positive relationship between the shareholding pledge of the controlling shareholders and the financing of enterprises.

**The impact of senior executives' overseas experience on the relationship between controlling shareholders' equity pledge and corporate excessive financialization.** The overseas experience of senior executives may have two different effects on the relationship between equity pledge of controlling shareholders and excessive financialization of enterprises. On the one hand, executives with overseas experience pay more attention to research and development, which can promote enterprise technological innovation [44]. Due to the limited assets owned by the enterprise, the decision-making of the enterprise is more inclined to the entity investment and R&D expenditure related to the company's business, which will correspondingly reduce the investment in financial assets. Therefore, senior executives with overseas experience may alleviate the positive relationship between equity pledge of controlling shareholders and corporate financialization.

On the other hand, based on the social capital theory, the overseas experience of senior executives will promote the relationship between the equity pledge of controlling shareholders and the over-financing of enterprises. The social capital of executives refers to the ability of executives to establish social networks and obtain information, knowledge and other resources from them. Its specific manifestations include political relations, employment networks, alumni relations and other forms. The higher the benefits that executives can obtain from these relationships, the higher their social capital. Studies have shown that, in developing countries with weak legal and regulatory systems, social capital can directly improve the profitability of enterprises [45]. When executives with overseas experience study or work abroad, they miss the opportunity to establish contact with the local [46]. In the absence of local information and contact, executives with overseas background will face greater performance pressure [47]. At the same time, in order to avoid the risk of control transfer, the controlling shareholders of equity pledge will exert performance pressure on the management. Under the dual pressure, executives with overseas experience may choose to allocate more financial assets to improve short-term performance.

This paper examines the heterogeneity of executives according to whether they have overseas experience. The regression results are shown in Table 11. The group of senior executives with overseas experience is significantly positive at the level of 1%, while the group of senior executives without overseas experience is positive but not significant, indicating that the overseas experience of senior executives has promoted the relationship between the equity pledge of controlling shareholders and the excessive financialization of enterprises.

## Conclusions and suggestions

### Research conclusion

This paper takes non-financial listed companies in China from 2010 to 2020 as a research sample to explore the impact of equity pledge of controlling shareholders on corporate over-financialization and its internal impact mechanism, and draws the following conclusions. Firstly, the negative influence of equity pledge of controlling shareholders on the future main business performance of enterprises only exists in the sample of enterprises with excessive financialization. If the over-financialized enterprises have the behavior of equity pledge of controlling shareholders, it will significantly inhibit the development of the future main business performance of enterprises, while the moderately financialized enterprises have no significant impact on the future main business performance of enterprises. Secondly, equity pledge of controlling shareholders will aggravate the degree of over-financialization of enterprises, and the higher the proportion of equity pledge of controlling shareholders, the greater the degree of over-financialization of enterprises. Thirdly, the impact of equity pledge on the excessive financialization of enterprises is more prominent under the circumstances of low economic policy

**Table 11. Impact of overseas experience of senior executives.**

| Variables | With overseas experience | | Without overseas experience | |
|---|---|---|---|---|
| | (1) | (2) | (3) | (4) |
| | *Exfin* | *Exfin* | *Exfin* | *Exfin* |
| Pledge_d | 0.0048** | | 0.0016 | |
| | (2.3666) | | (0.7906) | |
| Pledge_r | | 0.0078*** | | 0.0052 |
| | | (2.6536) | | (1.4928) |
| $Fin_{i,t-1}$ | -0.3898*** | -0.3895*** | -0.4148*** | -0.4154*** |
| | (-20.3210) | (-20.2910) | (-16.8321) | (-16.8594) |
| Lev | 0.0059 | 0.0059 | -0.0127 | -0.0133 |
| | (0.7006) | (0.6998) | (-1.2581) | (-1.3296) |
| Cf | 0.0237** | 0.0238** | 0.0240** | 0.0239** |
| | (2.2237) | (2.2386) | (2.0169) | (2.0060) |
| Size | -0.0065** | -0.0064** | -0.0061** | -0.0061** |
| | (-2.4720) | (-2.4393) | (-2.2140) | (-2.2310) |
| Roa | -0.0261* | -0.0259* | -0.0594*** | -0.0596*** |
| | (-1.9233) | (-1.9082) | (-3.5458) | (-3.5604) |
| Top1 | -0.0001 | -0.0001 | 0.0001 | 0.0001 |
| | (-1.0149) | (-0.9975) | (0.3693) | (0.3884) |
| Assetturn | -0.0051 | -0.0050 | -0.0057 | -0.0057 |
| | (-1.1625) | (-1.1396) | (-1.4103) | (-1.3989) |
| Fixed | -0.0723*** | -0.0721*** | -0.0394*** | -0.0390*** |
| | (-6.4877) | (-6.4755) | (-3.3427) | (-3.3171) |
| TobinQ | 0.0021*** | 0.0022*** | 0.0013 | 0.0013 |
| | (2.6811) | (2.7589) | (1.0664) | (1.0404) |
| Duality | 0.0016 | 0.0016 | 0.0016 | 0.0015 |
| | (0.7259) | (0.7468) | (0.5936) | (0.5724) |
| Pay3 | 0.0001 | 0.0002 | 0.0021 | 0.0022 |
| | (0.0746) | (0.0804) | (0.8600) | (0.8791) |
| Indep | -0.0001 | -0.0001 | 0.0001 | 0.0001 |
| | (-0.4042) | (-0.3925) | (0.2971) | (0.2875) |
| Constant | 0.1745*** | 0.1719*** | 0.1399** | 0.1399** |
| | (2.7447) | (2.7076) | (2.0252) | (2.0285) |
| Year Fixed effect | YES | YES | YES | YES |
| Corporate fixed effect | YES | YES | YES | YES |
| N | 11903 | 11903 | 8999 | 8999 |
| R2_a | 0.1548 | 0.1550 | 0.1779 | 0.1782 |

Note: This table shows the results of the tests grouped by executives' overseas experience. The definitions of the variables are provided in Table 1. Firm and year fixed effects are included in the models. The coefficient estimates and t-statistics are reported based on robust standard errors clustered by firm.

*** $p < 0.01$,

** $p < 0.05$,

* $p < 0.1$; in parentheses are the t-statistics of the corresponding coefficients.

uncertainty, less fierce industry competition and overseas experience of enterprise executives. This is because in China, enterprises increase investment in financial assets to chase high profits in the financial market, while the uncertainty of economic policies intensifies the volatility of the financial market. The fierce competition in the product market restrains the short-term

opportunistic behavior of the management to some extent, and then restrains the degree of excessive financialization of enterprises with controlling shareholders' equity pledge. Executives with overseas experience will face greater performance pressure and increase the degree of excessive financialization of controlling shareholder equity pledge enterprises. Finally, the management's short-sighted behavior caused by performance pressure is the key factor for controlling shareholder equity pledge enterprises to adjust their investment strategies. Equity pledge by controlling shareholders has a negative impact on the performance of listed companies, enhances the short-sighted decisions of management, and further increases the degree of excessive financialization of enterprises.

## Policy recommendations

The following policy recommendations can be obtained from the above research conclusions:

Firstly, listed companies should pay attention to the construction of internal control system process. Listed companies can strengthen the sensitivity between the management's compensation and the company's main business performance, so as to reduce the management's short-term ism and prevent enterprises from changing their investment decisions due to the risk of stock price collapse, which will damage the company's long-term development.

Secondly, regulators should improve the equity pledge system and strengthen the disclosure of equity pledge information. The regulatory authorities should issue relevant information disclosure guidelines, establish a dynamic regulatory mechanism for controlling shareholder equity pledge companies in advance, during and after the event, and focus on enterprises with weak industry competition and enterprises with overseas experience of senior executives. Enterprises with equity pledge of controlling shareholders should timely and accurately disclose the investment direction of the funds, so that regulators, small and medium-sized shareholders and investors can understand the use direction of the funds in a timely manner, and ensure that the use of funds is legal and compliant, thus reducing the over-financing problem caused by the individual equity pledge of controlling shareholders to a certain extent.

Finally, further promote industrial upgrading. The government should create a good industrial investment atmosphere, guide real enterprises to develop their main businesses, improve the rate of return of real investment through industrial upgrading, and reduce the possibility of enterprises increasing financial investment in order to pursue profits.

## Supporting information

**S1 Data.**
(DTA)

**S2 Data.**
(DO)

## Author Contributions

**Conceptualization:** Yue Xie, Tianhui Wang.

**Data curation:** Tianhui Wang.

**Funding acquisition:** Yue Xie, Jinhua Zhang.

**Project administration:** Yue Xie, Jinhua Zhang.

**Supervision:** Yue Xie.

**Validation:** Yue Xie.

**Writing – original draft:** Tianhui Wang.

**Writing – review & editing:** Yue Xie, Tianhui Wang, Na Wang.

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
