## [Decision Letter · Decision Letter 0]

23 May 2023

PONE-D-23-11287Does Controlling Shareholders' Share Pledge Exacerbate Excessive Financialization of Enter-prises? ——Evidence from Performance Pressure PerspectivePLOS ONE

Dear Dr. Wang,

Thank you for submitting your manuscript to PLOS ONE. After careful consideration, we feel that it has merit but does not fully meet PLOS ONE’s publication criteria as it currently stands. Therefore, we invite you to submit a revised version of the manuscript that addresses the points raised during the review process.

ACADEMIC EDITOR:The authors need to link their findings more strongly to context, highlight their economic, academic/research and policy/practice implications.I may suggest the authors add the discussion of the findings in the Chinese context. How their findings are different from other studies? What can we learn from the Chinese context?==============================

We look forward to receiving your revised manuscript.

Kind regards,

Chenguel Mohamed Bechir, Phd HDR

Academic Editor

PLOS ONE

Journal Requirements:

Additional Editor Comments:

The authors need to link their findings more strongly to context, highlight their economic, academic/research and policy/practice implications.

I may suggest the authors add the discussion of the findings in the Chinese context. How their findings are different from other studies? What can we learn from the Chinese context?

Reviewers' comments:

Reviewer's Responses to Questions

**Comments to the Author**

1. Is the manuscript technically sound, and do the data support the conclusions?

Reviewer #1: Yes

Reviewer #2: Yes

2. Has the statistical analysis been performed appropriately and rigorously? 

Reviewer #1: Yes

Reviewer #2: Yes

3. Have the authors made all data underlying the findings in their manuscript fully available?

Reviewer #1: Yes

Reviewer #2: Yes

4. Is the manuscript presented in an intelligible fashion and written in standard English?

Reviewer #1: Yes

Reviewer #2: No

5. Review Comments to the Author

Reviewer #1: The authors need to link their findings more strongly to context, highlight their economic, academic/research and policy/practice implications.

I may suggest the authors add the discussion of the findings in the Chinese context. How their findings are different from other studies? What can we learn from the Chinese context?

Reviewer #2: An interesting topic, check for grammar and conjugation errors and add the article at the end of the introduction

The database is recent and the analyses are very impressive

I congratulate the authors for this article

6. PLOS authors have the option to publish the peer review history of their article (what does this mean?). If published, this will include your full peer review and any attached files.

Reviewer #1: **Yes: **Moncef Guizani

Reviewer #2: No

---

## [Author Response · Author response to Decision Letter 0]

24 Jun 2023

We would like to thank the referees and the editor for their helpful comments and constructive suggestions. We have revised the paper accordingly, the major changes are marked in blue. In the following, we give our responses to the reviewer. And we mention the changes made in the revised manuscript. All of the major changes are marked in blue in the revised version.

Response to the Academic Editor:

Editor’s additional comments: 

1. After careful consideration, we feel that it has merit but does not fully meet PLOS ONE’s publication criteria as it currently stands. Therefore, we invite you to submit a revised version of the manuscript that addresses the points raised during the review process.

Authors’ Response: The authors would like to thank the Editor for the comment. We have addressed the comments of the Reviewer. In the following, we give our responses to the comments of the Reviewer.

2. The authors need to link their findings more strongly to context, highlight their economic, academic/research and policy/practice implications.

I may suggest the authors add the discussion of the findings in the Chinese context. How their findings are different from other studies? What can we learn from the Chinese context?

Authors’ Response: Thanks for the comment. We have added the research significance in the introduction in the revised manuscript. At the same time, we have revised the expression of contribution at the end of the introduction and the expression of the conclusion in the revised manuscript.

Response to the Editor:

Journal Requirements:

Authors’ Response: Thanks for reminding me. I have modified the paper format according to the journal template. If the format needs to be modified, please inform me again by email. Thank you.

Authors’ Response: Thanks for reminding me. The correct information is as follows:

Funding: This research was funded by the National Natural Science Foundation of China (grant number: 72002201), URL of funder website: https://www.nsfc.gov.cn/; And the National Social Science Fund of China (grant number: 21BJY256), URL of funder website: http://fz.people.com.cn/skygb/sk/index.php/index/seach/

Authors’ Response: Thanks for reminding me. After careful examination, the references we quoted are complete and correct. We did not cite the retracted paper. In the contribution at the end of the introduction, we add the literature that is more similar to our study, namely 5-7 in the references. Except for the three new references, we have not changed the rest of the references.

Response to the Reviewer #1

The reviewer has the following comments:

1. The authors need to link their findings more strongly to context, highlight their economic, academic/research and policy/practice implications.

Authors’ Response: Thanks for the comment. We have added the research significance in the introduction in the revised manuscript.

2. I may suggest the authors add the discussion of the findings in the Chinese context. 

Authors’ Response: Thanks for this suggestion. We have revised the expression of the conclusion in the revised manuscript.

3. How their findings are different from other studies? What can we learn from the Chinese context?

Authors’ Response: Thanks for this suggestion. We have revised the expression of contribution at the end of the introduction in the revised manuscript.

Response to the Reviewer #2

The reviewer has the following comments:

1. An interesting topic, check for grammar and conjugation errors and add the article at the end of the introduction.

The database is recent and the analyses are very impressive.

I congratulate the authors for this article.

Authors’ Response: Thank you for your suggestions and affirmation of our paper. We have revised the grammar and conjugation errors in the revised manuscript. At the same time, we have added some articles similar to our paper at the end of the introduction in the revised manuscript.

 

We tried our best to improve the manuscript and made some changes in the manuscript. These changes will not influence the content and framework of the paper. And here we did not list all the changes but marked in blue in revised paper.

We appreciate for Editors/Reviewers’ warm work earnestly, and hope that the correction will meet with approval.

Once again, thank you very much for your comments and suggestions.

---

## [Editor Report · Decision Letter 1]

4 Jul 2023

Does Controlling Shareholders' Share Pledge Exacerbate Excessive Financialization of Enterprises? ——Evidence from Performance Pressure Perspective

PONE-D-23-11287R1

Dear Dr. Tianhui Wang

We’re pleased to inform you that your manuscript has been judged scientifically suitable for publication and will be formally accepted for publication once it meets all outstanding technical requirements.

Kind regards,

Chenguel Mohamed Bechir, Phd HDR

Academic Editor

PLOS ONE

Additional Editor Comments (optional):

thank you for your corrections and response
---

## [Editor Report · Acceptance letter]

10 Jul 2023

PONE-D-23-11287R1 

Does controlling shareholders' share pledge exacerbate excessive financialization of enterprises? —Evidence from performance pressure perspective 

Dear Dr. Wang:

I'm pleased to inform you that your manuscript has been deemed suitable for publication in PLOS ONE. Congratulations! Your manuscript is now with our production department. 

Kind regards, 

on behalf of

Dr. Chenguel Mohamed Bechir 

Academic Editor

PLOS ONE